# Should We Use High-Flow Nasal Cannula in Patients Receiving Gastrointestinal Endoscopies? Critical Appraisals through Updated Meta-Analyses with Multiple Methodologies and Depiction of Certainty of Evidence

**DOI:** 10.3390/jcm11133860

**Published:** 2022-07-03

**Authors:** Chi Chan Lee, Teressa Reanne Ju, Pei Chun Lai, Hsin-Ti Lin, Yen Ta Huang

**Affiliations:** 1Department of Critical Care Medicine, Guam Regional Medicine City, Dededo, GU 96913, USA; ccl1985@gwu.edu; 2Department of Internal Medicine, New York Presbyterian Queens, Flushing, NY 11355, USA; tej9016@nyp.org; 3Education Center, National Cheng Kung University Hospital, College of Medicine, National Cheng Kung University, Tainan 70403, Taiwan; debbie0613.lai@gmail.com; 4Department of Medicine, Case Western Reserve University and MetroHealth Medical Center, Cleveland, OH 44109, USA; ccindylin@gmail.com; 5Department of Surgery, National Cheng Kung University Hospital, College of Medicine, National Cheng Kung University, Tainan 70403, Taiwan

**Keywords:** high-flow nasal cannula, high flow nasal cannula (HFNC), EGD, ERCP, gastrointestinal endoscopy, sedation, monitored anesthesia care, oxygen therapy

## Abstract

(1) Background: High-flow nasal cannula (HFNC) therapy or conventional oxygen therapy (COT) are typically applied during gastrointestinal (GI) endoscopic sedation. (2) Methods: We conducted a rigorous systematic review enrolling randomized controlled trials (RCTs) from five databases. Risk of bias was assessed using Cochrane’s RoB 2.0 tool; certainty of evidence (CoE) was assessed using GRADE framework. Meta-analysis was conducted using inverse-variance heterogeneity model and presented as relative risk (RR) with 95% confidence interval (CI). Trial sequential analysis was performed, and sensitivity analysis was conducted with Bayesian approach. (3) Results: Eight RCTs were included. Compared to COT, HFNC did not reduce the overall incidence of hypoxemia (RR 0.51; 95% CI 0.24–1.09; CoE: very low) but might reduce the incidence of hypoxemia in patients at moderate to high risk for hypoxemia (RR 0.54; 95% CI 0.31–0.96; and CoE: very low). HFNC might reduce the incidence of severe hypoxemia (RR 0.38; 95% CI 0.20–0.74; and CoE: low). HFNC might not affect the need of minor airway interventions (RR 0.31; 95% CI 0.08–1.22; and CoE: very low) and had no effect on procedure duration (CoE: very low); (4) Conclusions: During GI endoscopic sedation, HFNC might reduce the incidence of hypoxemia in patients at moderate to high risk for hypoxemia and prevent severe hypoxemia.

## 1. Introduction

Endoscopic procedures are common procedures performed worldwide for the diagnosis and treatment of gastrointestinal (GI) disorders. In 2019, there were a total of 22.2 million GI endoscopies performed in the United States, of which 64% were colonoscopies and 34% were esophagogastroduodenoscopies (EGD) [1]. While GI endoscopies are generally considered to be safe, patients’ willingness to receive procedures could be limited due to anxiety and peri-procedural discomfort. For that reason, administering intravenous (IV) sedatives during GI endoscopies has become a common approach for gastroenterologists in U.S. [2]. A nationwide survey in the U.S. showed that more than 98% of endoscopies were performed under sedation [3]. Although sedatives could improve patients’ satisfaction, they may have deleterious effects on hemodynamic profiles and respiratory functions. Incidence of hypoxemia during endoscopic sedation largely varies, based on the types of procedures and population. In the elder population receiving endoscopic retrograde cholangiopancreatography (ERCP), the incidence of hypoxemia could be as high as 21.4% [4]. Prolonged hypoxemia during sedation is a major risk factor of cardiac arrhythmia or myocardial infarction [5].

Conventional oxygen therapies (COT), such as low-flow nasal cannula, modified face mask, and nasopharyngeal airway, have been used to prevent hypoxemia during endoscopic sedation [6]. Nevertheless, the capability to provide a high fraction of inspired oxygen concentrations (FiO_2_) through those devices is limited because their low-flow rates may not meet patients’ peak inspiration flow, resulting in oxygen mixing with carbon dioxide (CO_2_) in the dead space. High-flow nasal cannula (HFNC) is an innovative high-flow oxygen-delivery system that can deliver humified oxygen up to 60 L per minute. HFNC not only makes higher FiO_2_ possible by providing a gas-flow rate which meets a patient’s peak inspiratory flow but also decreases the work of breathing by washing out CO_2_ in the dead space [7]. High-flow gas rates may generate small positive end-expiratory pressure (PEEP) by creating tracheal gas insufflation which prevents atelectasis [7]. HFNC has been studied in patients receiving bronchoscopy [8], awake craniotomy [9], and dental procedures [10] under IV sedation.

Several randomized controlled trials (RCTs) [11,12,13,14,15,16,17,18] had studied the use of HFNC in patients who received GI endoscopies during sedation. However, the study designs were heterogeneous, and study results were inconsistent. Four recent systematic reviews (SRs) were published to demonstrate the benefit in use of HFNC for prevention of hypoxemia in patients who received GI endoscopies [19,20,21,22]. Based on the tool of “A Measurement Tool to Assess Systematic Reviews II” (AMSTAR2) [23], these SRs were appraised as critically low or of low quality, mainly due to lack of discussing the influence of RoB on the pooled estimates (Table A1). Besides, certainty of evidence (CoE) in each outcome was not rated in these SRs. An additional RCT [18] was recently published on the same topic. Therefore, the present study aims to synthesize the updated evidence of using HFNC in patients who received GI endoscopies under IV sedation through comprehensive SR, multiple advanced methodologies in statistics, and rating the CoE.

## 2. Materials and Methods

### 2.1. Protocol and Registration

Two independent reviewers (CCL and HTL) performed a comprehensive search for relevant articles in multiple databases, including MEDLINE, Embase, Cochrane library, and Airiti library. We also searched ClinicalTrials.gov Database for unpublished or ongoing trials. No language limitation was applied. Relevant citations originating from references were eligible for additional review and selection. We conducted searches of electronic databases, both with controlled vocabulary (MeSH/EMTREE) terms and free text terms using the following keywords: (‘high flow nasal cannula’ OR ‘HFNC’ OR ‘high flow oxygen therapy’ OR ‘oxygen therapy’ OR ‘HFNO’ OR ‘high flow nasal prong’ OR ‘high flow nasal oxygenation’) AND (‘EGD’ OR ‘endoscope’ OR ‘ERCP’ OR ‘esophagogastroduodenoscopy’ OR ‘colonoscope’ OR ‘endoscopy’ OR ‘colonoscopy’ OR ‘endoscopic retrograde cholangiopancreatography’). The last search was conducted on 11 June 2022. The protocol for this SR was registered on PROSPERO (CRD42021272313) and designed according to Cochrane Handbook for Systematic Reviews of Interventions [24] and Preferred Reporting Items for Systematic Reviews and Meta-Analysis (PRISMA) guideline.

### 2.2. Study Selection

To avoid confounders and selection bias in non-RCTs, we only included studies that were designed as prospective RCTs. The inclusion criteria were: (1) participants who received GI endoscopies under procedural sedation, (2) participants who were randomly assigned to receive HFNC for oxygen therapy in the intervention group and receive COT (low-flow nasal cannula, nasal prong, mouthguard, or nasopharyngeal catheter) for oxygen therapy in the control group, and (3) studies which reported any of the following outcomes: incidence of hypoxemia, severe hypoxemia, hypercapnia, minor airway interventions (chin lift, jaw thrust, or insertion of nasopharyngeal airway), or duration of procedures. Studies were excluded if they were abstracts, conference articles, trial protocols, or performed in the pediatric population (age < 16 years old). 

### 2.3. Data Extraction

Two independent reviewers (CCL and HTL) extracted the following data from selected articles: characteristics of the study (first author, year of publication, country, and sample size); characteristics of the participants (age, body mass index, and underlying medical conditions); characteristics of the procedures (types of procedures, sedation protocols, HFNC settings, COT settings); and study outcomes mentioned in Section 2.2. Incidence of hypoxemia, severe hypoxemia, or hypercapnia indicated the numbers of events when patients’ oxygen saturation by pulse oximetry (SpO_2_) or CO_2_ levels were below, or above, certain levels defined by each study. Minor airway interventions included chin lift, jaw thrust, or nasal airway insertion. If the incidences of minor airway interventions were reported separately, we extracted the incidence of the most common intervention for meta-analysis. Discrepancies of data collection were resolved in consultation with a third reviewer (TJ). We contacted primary investigators through email if additional information was required for analysis. Missing data were mentioned in the results section and excluded from data analysis. 

### 2.4. Quality Assessment

The quality of included studies was assessed by two reviewers (CCL and HTL) independently and discrepancies were resolved in consultation with a third reviewer (TJ). The risk of bias (RoB) of studies was assessed according to Cochrane’s risk-of-bias tool for randomized trials, version 2.0 (RoB 2) [25]. Cochrane’s RoB 2 tool is structured to assess study bias in five domains. RoB judgments on each domain can be made as “low risk”, “some concerns” or “high risk”. An overall RoB judgment will be made based on the judgment of five domains. 

### 2.5. Study Analysis

#### 2.5.1. Meta-Analysis with Inverse Variance Heterogeneity Model

In contrast to the Mantel–Haenszel (M–H) model utilized by the previous meta-analysis [19], we conducted a meta-analysis using Microsoft Excel (Microsoft, Redmont, WA, USA) add-in MetaXL 5.3 (EpiGear International, Sunrise Beach, Australia) with the inverse variance heterogeneity (IVhet) model [26]. IVhet model, compared to random-effects (RE) model, might retain a correct coverage probability and lower observed variance than the RE model estimator, regardless of heterogeneity [26,27]. Risk ratio (RR) with 95% confidence level (CI) was reported for dichotomous outcome, and weighted mean difference (WMD) with 95% CI was reported for continuous outcome. For the study conducted by Teng et al. [12], we combined the nasal cannula group and mandibular advancement bite block group into one single control group, according to the Cochrane handbook, version 5.1 Chapter 16.5 [28]. A two-sided *p* value < 0.05 was considered statistically significant. Heterogeneity among studies was assessed using the *I*^2^ tests. An *I*^2^ higher than 50% represented substantial heterogeneity. Predefined subgroup analysis was conducted to compare studies which enrolled patients at low risk for hypoxemia (e.g., American Society of Anesthesiologists (ASA) class I–II) versus those at moderate to high risk for hypoxemia due to patients’ health conditions (e.g., obesity, higher ASA score, and history of sleep apnea) or requirements of complex procedures (e.g., ERCP). In addition, we performed subgroup analysis based on RoB among studies (i.e., studies with high overall RoB versus those with low or some concerns of overall RoB).

#### 2.5.2. Sensitive Analysis

To better handle the zero events, we conducted a sensitive analysis with a Bayesian approach by utilizing the interactive web-based tool MetaInsight (https://crsu.shinyapps.io/metainsightc accessed on 16 March 2022). MetaInsight [29], a free and recently developed software created based on the existing *netmeta* and *shiny* packages for R, allows users to conduct Bayesian meta-analysis without the requirement of advanced statistical programming expertise. RR with 95% credible interval (CrI) was reported for dichotomous outcome, and WMD with 95% CrI was reported for continuous outcome. Bayesian RE meta-analysis naturally allows the presence of imprecision in the estimated between-study variance and carries better compatibility with studies including zero events [30]. Compared to other Frequentist approaches, a Bayesian approach likely performs better in case of a small number of studies (N < 5) or the presence of zero events [31].

#### 2.5.3. Trial Sequential Analysis

To avoid the risk of type I and type II errors resulting from sequential testing on a constant significance level, we conducted a trial sequential analysis (TSA) using TSA version 0.9.5.10 beta (Copenhagen Trial Unit, Center for Clinical Intervention Research, Rigshospitalet, Copenhagen, Denmark). Types I and II errors were set at 5 and 20%, respectively. A RE model with Biggerstaff–Tweedie (BT) method was used. For zero events, a constant addition of 0.001 was chosen in the software. O’Brien–Fleming α-spending monitoring boundaries were applied for hypothesis testing.

Results were considered true positive if the Z curve crossed the O’Brien–Fleming monitoring boundaries and considered true negative if the Z curve entered the futility area. An underpower was detected if total sample size of included studies did not achieve the required information size (RIS). The RIS was calculated considering the proportion of investigational and control events and the anticipated heterogeneity variance of the meta-analysis. The incidence of intervention and control arms was filled in the “overall events/total cases” of the enrolled studies.

### 2.6. CoE

The CoE of outcomes was assessed using the Grading of Recommendations Assessment, Development and Evaluation (GRADE) framework [32]. The overall CoE was judged by five downgrading domains for the SR of RCTs. We downgraded CoE in the domain of RoB based on the proportion of RCTs with some-concern and/or high overall RoB. In the domain of inconsistency, we downgraded one or two levels if *I^2^* ratio was more than 50% or 90%, respectively. If inappropriate combination in population/intervention/comparator/outcome was noted, we downgraded level(s) in the domain of indirectness. Apart from wide ranges of interval, probability of false positive or negative and insufficient sample size were evaluated in the domain of imprecision. We considered the major asymmetry of Doi plot as probability of publication bias. Disagreements of GRADE assessments were discussed and resolved in the groups.

## 3. Results

### 3.1. Summary of the Characteristics of Studies

Figure 1 showed the study selection process. Overall, we identified 2593 articles from Medline, Embase, Cochrane library, Airiti library and ClinicalTrials.gov. After 234 duplicate articles were removed, we screened 2359 articles. A total of 13 full-text articles which met the inclusion criteria were assessed for eligibility. Five articles were not enrolled since they were study protocols, conference abstracts, or study performed in pediatric population. Finally, we included eight RCTs [11,12,13,14,15,16,17,18] involving a total of 3236 patients in the analysis.

Table 1 showed study characteristics. Two studies included patients who received EGD and were at low risk for hypoxemia. Six studies included patients who received EGD, EGD and interventions, ERCP, or colonoscopy and were at moderate to high risk for hypoxemia. Two studies received academic funding and four studies received industrial as well as academic funding. Participants’ demographic profiles were heterogenous across studies. Two studies did not specify sedation protocols and allowed physicians to administer sedatives to reach targeted level of sedation at their discretion. In terms of the initial choice of sedatives: one study used midazolam plus alfentanil IV push; one study used propofol with targeted controlled infusion (TCI); and other studies used propofol with or without fentanyl IV push. In terms of choice of sedatives to maintain targeted levels of sedation: two studies used propofol IV push as needed; two studies used propofol IV continuous infusion; one study used either propofol IV push or continuous infusion; and one study used propofol TCI. Depth of sedation ranged from moderate sedation to general anesthesia. All but one study [12] reported the significant difference of total dose of sedatives or analgesics between HFNC group and COT group. Most studies used Optiflow as their HFNC devices. Regarding HFNC settings, three studies [13,16,18] matched FiO_2_ in HFNC groups to those in control groups, while other studies used 100% of FiO_2_. Flow of oxygen in HFNC groups ranged from 20 to 70 L per minute. In contrast, flow of oxygen in COT groups ranged from 2–6 L per minute.

### 3.2. RoB Accessment

Figure 2 showed the study bias of included studies. Two studies did not detail the methods of allocation concealment. Although blinding of participants or physicians cannot be avoided in all studies due to different appearances of oxygen devices, no deviations from the intended interventions were observed. Missing outcome data was not observed during the short duration of study period. Four studies did not detail the methods of blinding to outcome assessors. Only one study did not detail the pre-specified analysis plan. After appraisal, two studies were deemed high overall RoB due to some concerns of bias in two different domains. Only three RCTs fit the criteria of low overall RoB.

### 3.3. Study Outcome

#### 3.3.1. Incidence of Hypoxemia

All eight studies reported incidence of hypoxemia. Overall, meta-analysis conducted with IVhet model showed HFNC did not reduce the incidence of hypoxemia in comparison with COT (RR 0.51; 95% CI 0.24–1.09; Figure 3a). Significant heterogeneities among studies were observed (*I*^2^ = 71%). TSA depicted a fluctuated Z-curve, and the end of Z-curve lies on the O’Brien–Fleming monitoring boundaries, indicating the possibility of an uncertain result (Figure 3b). Besides, TSA also showed that studies were underpowered since the total sample size of included studies did not achieve RIS (Figure 3b). Doi plot showed major asymmetry (Figure A1a), and CoE was rated as very low which was downgraded in the domains of RoB, inconsistency, imprecision, and publication bias (Table 2). Similarly, when considering the three RCTs with low overall RoB, non-significant results were observed (RR 0.56; 95% CI 0.28–1.11; *I*^2^ = 77%; Figure A2). Although the end of the Z-curve crossed the O’Brien–Fleming α-spending monitoring boundaries, it did not reach the line of RIS which indicated the existence of uncertainty (Figure A3).

Subgroup analysis (Figure 3a) showed HFNC might not reduce the incidence of hypoxemia in patients at low risk for hypoxemia with a very wide range of interval (RR 0.05; 95% CI 0.00–1.07; *I*^2^ = 69%). However, HFNC may reduce the incidence of hypoxemia in patients at moderate to high risk for hypoxemia (RR 0.54; 95% CI 0.31–0.96; and *I*^2^ = 59%). TSA of studies which only included patients at moderate to high risk for hypoxemia yielded a true positive result since the Z-line crossed the O’Brien–Fleming monitoring boundaries and exceeded the line of RIS (Figure 3c). CoE for incidence of hypoxemia in patients at moderate to high risk for hypoxemia was rated as very low based on downgrading in the domains of RoB, inconsistency, and publication bias (Table 2).

Bayesian meta-analysis (Figure A4), in contrast to the analysis conducted with IVhet model, showed that HFNC might reduce the incidence of hypoxemia in comparison with COT (RR 0.155; 95% CrI 0.014–0.862). Subgroup analysis showed HFNC may not reduce the incidence of hypoxemia in patients at low risk for hypoxemia (RR 0.00569; 95% CrI 0.00–1.04) but may reduce the incidence of hypoxemia for patients at moderate to high risk for hypoxemia (RR 0.404; 95% CrI 0.12–0.971).

#### 3.3.2. Incidence of Severe Hypoxemia

Only four studies reported the incidence of severe hypoxemia. Severe hypoxemia was defined as SpO_2_ < 80% in two studies, <85% in one study, and <75% in one study. Meta-analysis conducted with IVhet model revealed HFNC might reduce the incidence of severe hypoxemia (RR 0.38; 95% CI 0.20–0.74; Figure 4a) in comparison with COT. TSA of all four studies showed a true positive effect with sufficient sample sizes but fluctuated curve (Figure 4b). CoE was rated as low because the domains of RoB and publication bias (Figure A1b) were downgraded (Table 2). Subgroup analysis showed HFNC may reduce the incidence of hypoxemia for patients at moderate to high risk for hypoxemia (RR 0.42; 95% CI 0.21–0.83). However, TSA of studies including patients at moderate to high risk for hypoxemia may be false positive and underpowered (Figure 4c).

Bayesian meta-analysis (Figure A5) also showed HFNC might reduce the incidence of hypoxemia in comparison with COT (RR 0.177; 95% CrI 0.014–0.806). However, subgroup analysis showed HFNC might not reduce the incidence of severe hypoxemia for patients at moderate to high risk for hypoxemia (RR 0.3; 95% CrI 0.0493–1.12).

#### 3.3.3. Incidence of Hypercapnia

Mazzeffi et al. and Thiruvenkatarajan et al. assessed incidence of hypercapnia with transcutaneous blood carbon dioxide (P_t_CO_2_) measuring device and used P_t_CO_2_ > 20 mmHg as the cutoff value to diagnose hypercapnia. Meta-analysis depicted no statistical difference (RR 1.24; 95% CI 0.97–1.58; *I*^2^ = 0%; Figure A6). The end of Z-curve in TSA did not reach futility area, indicating the possibility of false negative (Figure A7). Reasonably, the Z-curve did not cross line of RIS in a pooling of just two RCTs, so more studies are warranted to give a more solid conclusion in this outcome. CoE was rated as low because the domain of imprecision was downgraded by two levels (Table 2).

#### 3.3.4. Need for Minor Airway Interventions

Six studies reported the need for minor airway interventions (e.g., jaw thrust, chin lift, or insertion of nasal airway). Although the need for minor airway interventions was lower in the HFNC group in comparison with COT (RR 0.31; Figure 5a), a very wide range crossing the non-significant line was noted (95% CI (0.08–1.22)). High heterogeneity was observed (*I*^2^ = 91%). Doi plot depicted minor asymmetry (Figure A1c). CoE was rated as very low (Table 2). In contrast, TSA depicted an absolute benefit of HFNC from the first study, so the O’Brien–Fleming monitoring boundaries and the line of RIS were not renderable (Figure 5b). Bayesian meta-analysis (Figure A8) showed that HFNC reduced the need for minor airway interventions in comparison with COT (RR 0.178; 95% CrI 0.0256–0.919).

#### 3.3.5. Duration of Procedure

There was no significant difference in the procedure time between HFNC group and COT group (WMD 0.12 min; 95% CI −0.04 to 0.28; *I*^2^ = 0%; Figure A9). TSA showed the pooled estimates may be underpowered and existed the probability of false negative (Figure A10). Besides, major asymmetry of Doi plot was observed (Figure A1d). CoE was rated as very low (Table 2). Bayesian meta-analysis also showed there was no significant difference in mean difference (MD) of procedure time between the two groups (MD 0.173 min; 95% CrI −0.617 to 1.63; Figure A11).

## 4. Discussion

The present analysis investigated the clinical efficacy of HFNC therapy in patients who received IV sedation during GI endoscopies. Overall, HFNC therapy did not reduce the incidence of hypoxemia. However, HFNC therapy might reduce the incidence of hypoxemia in patients at moderate to high risk for hypoxemia and overall incidence of severe hypoxemia. The evidence was insufficient to suggest the impact of HFNC therapy on the incidence of hypercapnia. HFNC therapy might reduce the incidence of minor airway interventions. The duration of endoscopic procedures might not be different between the two groups.

In contrast to previous SRs, our analysis showed HFNC did not reduce the overall incidence of hypoxemia [19,20,21]. This phenomenon could be explained by the different statistical models used by each SR, especially when the included studies had heterogenous study designs or zero events (i.e., zero incidence in one or both study group) [33]. In our study, we utilized three different advanced statistical methodologies to compare the analysis results and minimize study bias. IVhet model, compared to the traditional model of meta-analysis, has been reported to minimize the possibility of under- or over-estimating the heterogeneity compared to traditional model of meta-analysis with sparse studies [26]. TSA are warranted for containment of the pooled estimates given the small numbers of studies. Due to the presence of zero events in some studies of HFNC group, sensitivity analysis using Bayesian approach was also used for duplicate confirmation [33,34]. Overall, we believe the current evidence was insufficient to make strong recommendations for the use of HFNC therapy routinely in cases who received GI endoscopies on IV sedatives. TSA in our study further revealed the results in most of the endpoints were unpowered, which highlighted the importance of including more studies to confirm the therapeutic effects. The discrepant pooled estimates between IVhet model and Bayesian framework could be explained by the applications of different methodologies and statistical models in the analysis, which further explained the uncertain benefits of HFNC on this issue.

It is worth mentioning that in patients who were at moderate to high risk for hypoxemia, the subgroup analysis conducted by three different methodologies universally suggested HFNC may be beneficial to prevent hypoxemia. This finding is more consistent with our real-world practice. Sedation for GI endoscopies, even for simple EGD, should be considered high-risk anesthetics for some cases [35]. Endoscopies, especially ERCP, frequently require deeper sedation to suppress gag, cough, and laryngospasm reflex. Sedatives that are administered to achieve targeted levels of sedation potentially lead to airway collapse, hypoventilation, or deleterious effect on hemodynamic profiles, especially in cases of obesity, higher ASA score, or history of sleep apnea. Upper endoscopies, placed in aerodigestive tracts, often causes partial obstruction of airway since their diameters frequently exceeds the cross-sectional area of oropharynx [36]. HFNC may be able to mitigate complications, such as hypoxemia, during GI endoscopic sedation in patients with higher anesthetic risks. The CoE based on the methodology of GRADE was only rated as very low, which indicated there was high likelihood that the actual effect would be substantially different from our study results. Future large prospective studies are needed to delineate the types of population that will benefit most from HFNC therapy in this setting.

The incidence of severe hypoxemia, typically defined as SpO_2_ less than 80%, may be more clinically relevant than the incidence of hypoxemia when comparing the efficacy of different oxygen therapies. Our analysis suggested that HFNC therapy might be beneficial in preventing severe hypoxemia in comparison with COT based on three statistical methodologies. The primary physiologic benefit of HFNC to prevent severe hypoxemia is the capability to maintain high FiO_2_. During anesthetic induction, a recent published meta-analysis of 14 studies showed HFNC was superior to COT in improving oxygenation and prolonging safe apnea time [37]. However, it is worth mentioning high gas flow may reduce the oxygen content by pulling nitrogen content from environmental air. The high gas flow passing through a constricted area could decrease the downstream pressure leading to a vacuum effect, a phenomenon also observed in venturi mask [38]. Wetsch et al. [39] performed a simulation study which used HFNC for apnea oxygenation. Interestingly, highest oxygen content was reached at a flow of 10 to 20 L per minutes. Higher flow resulted in a slight decrease in oxygen content. Further studies are needed to establish the gas flow of HFNC that can achieve optimal oxygen delivery in the setting of apnea oxygenation. HFNC could also provide PEEP at roughly 1 cm H_2_O for every 10 L per minute of gas flow by causing tracheal gas insufflation [40], but the effect may be reduced during GI endoscopies when patients open their mouths. Given current existing literature, the evidence was limited since only a few studies reported this outcome. In addition, the CoE was rated as low in GRADE methodology. More RCTs are warranted to confirm the benefit of HFNC on this endpoint.

The effect of HFNC therapy on ventilation is complex. Most studies suggested HFNC improved hypercapnia and work of breathing by washing out CO_2_ from dead space and improving thoracoabdominal synchrony [41,42]. On the other hand, unnecessary oxygen administration from HFNC, especially in patients with chronic obstructive pulmonary disease, could lead to hypercapnia due to the change of dead space and modulation of Haldane effect [43]. In our analysis, only two studies reported incidence of hypercapnia using PtCO2. While both studies showed a non-statistical increase in the incidence of hypercapnia in the HFNC group, it was too early to draw conclusions of HFNC’s effect on ventilation during endoscopic sedation. Our pooled estimates in three methodologies and the CoE in GRADE supported the above-mentioned conclusion. Since hypoventilation from sedative use is a major concern, further studies are needed to address the effect of HFNC therapy on CO_2_ change during sedation. End tidal CO_2_, while widely available, might not be the ideal tool to assess ventilation in this clinical setting since high-flow gas would severely dilute exhaled CO_2_, leading to a falsely low reading [44].

Minor airway interventions are typically used to correct partial airway closure resulting from sedation. Increased needs for minor airway interventions can lead to interruptions of endoscopies and prolong the duration of procedures. In our analysis, the meta-analysis conducted by IVhent model and Bayesian model showed discordant results. This was likely due to heterogeneity of study designs and different methodologies in handling zero events. The need for minor airway interventions, to a certain degree, is reflective of the severity of hypoventilation. Since most included studies did not apply CO_2_ monitoring devices, the need for minor interventions was mostly determined by low SpO_2_ level, a late feature of hypoventilation. We believe future evidence is needed to assess the impact of HFNC therapy on the incidence of minor airway interventions.

There was no overall difference in the pooled mean duration of endoscopic procedure between HFNC group and COT group. Frequent need for minor airway interventions could possibly prolong the duration of procedure. However, the average numbers of minor airway interventions needed for patients in each study were unclear and the incidence of minor airway interventions varied among included studies.

Strengths of this review includes a comprehensive literature search, pre-registered protocols at PROSPERO, compliance to the PRISMA guideline for systematic review, the use of both Frequentist and Bayesian approaches for meta-analyses, advanced application of TSA, and using GRADE methodology to assess the CoE. Limitation of this review includes the heterogeneity of study designs and small number of included studies which resulted in the uncertain conclusion. In addition, meta-regression cannot be performed to elucidate the relationship among heterogeneities due to the small number of enrolled RCTs.

## 5. Conclusions

During GI endoscopies, HFNC may not reduce the overall incidence of hypoxemia. However, HFNC may be beneficial in preventing hypoxemia in patients who were at moderate to high risk for hypoxemia and severe hypoxemia. HFNC might not affect the need of minor airway interventions and had no effect on procedure duration. Future large RCTs are needed to elucidate the population who will benefit most from HFNC therapy while balancing the costs of devices.

## Figures and Tables

**Figure 1 jcm-11-03860-f001:**
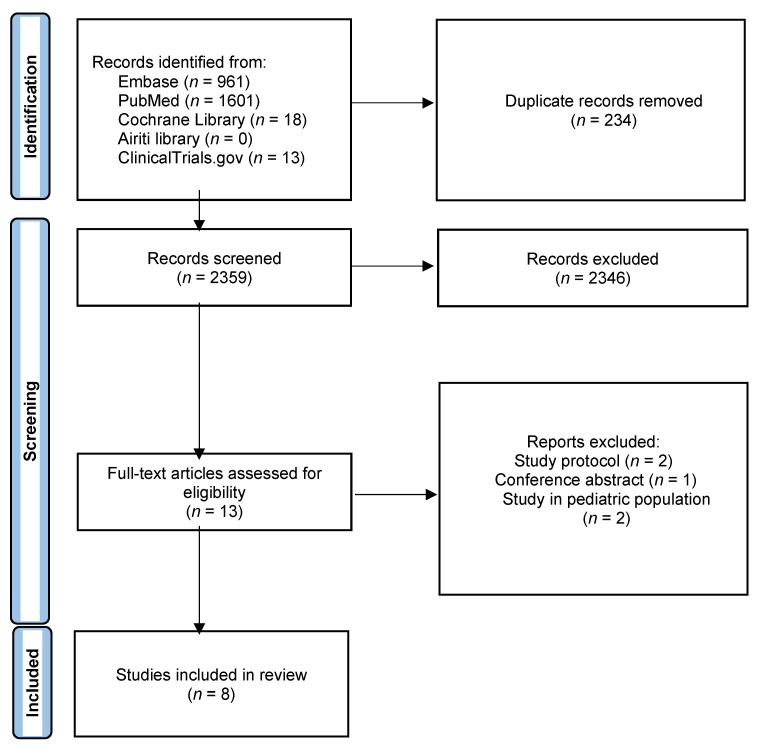
Study flow diagram.

**Figure 2 jcm-11-03860-f002:**
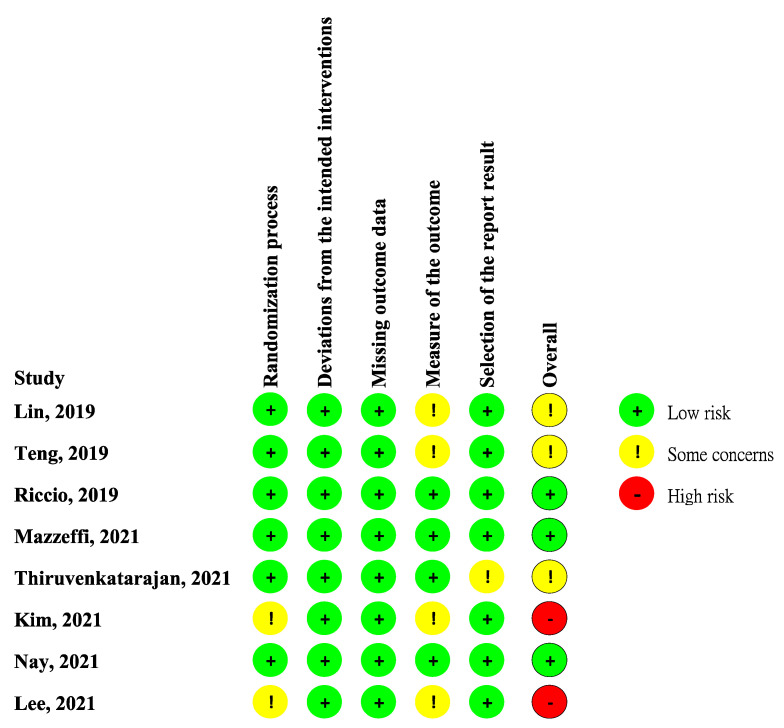
Risk of bias. Studies included Lin et al. [11], Teng et al. [12], Riccio et al. [13], Mazzeffi et al. [14], Thiruvenkataragan et al. [15], Kim et al. [17], Nay et al. [16], Lee et al. [18].

**Figure 3 jcm-11-03860-f003:**
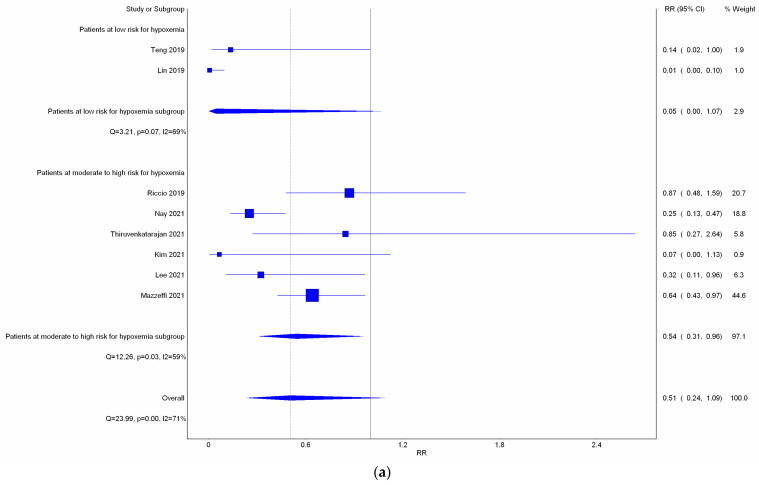
Incidence of hypoxemia. (**a**) Meta-analysis with IVhet model; (**b**) trial sequential analysis; and (**c**) trial sequential analysis: Patients with moderate to high risk for hypoxemia. Studies in Figure 3a include Riccio et al. [3], Nay et al. [16], Thiruyenkatarajan et al. [15], Kim et al. [17], Lee et al. [18], Mazzeffi et al. [14].

**Figure 4 jcm-11-03860-f004:**
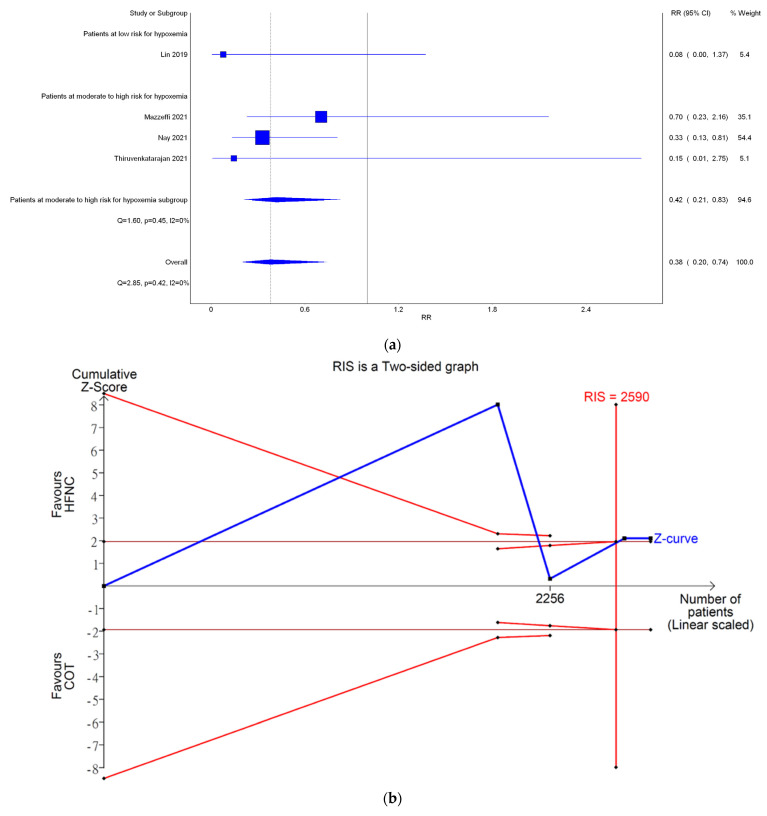
Incidence of severe hypoxemia. (**a**) Meta-analysis with IVhet model; (**b**) trial sequential analysis; and (**c**) trial sequential analysis: patients with moderate to high risk for hypoxemia. Studies in Figure 4a include: Mazzeffi et al. [14], Nay et al. [16], and Thiruvenkatarajan et al. [15].

**Figure 5 jcm-11-03860-f005:**
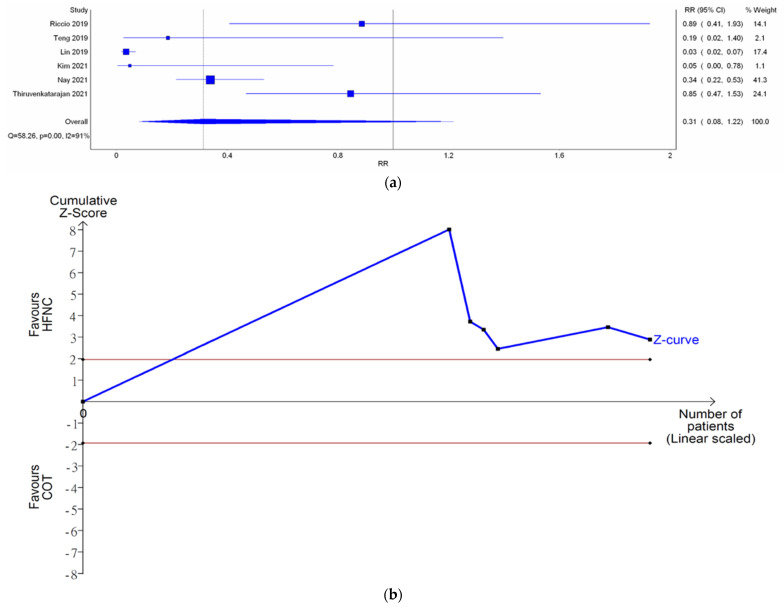
Need for minor airway interventions. (**a**) Meta-analysis with IVhet model; and (**b**) trial sequential analysis. Riccio et al. [13], Teng et al. [12], Lin et al. [11], Kim et al. [17], Nay et al. [16], Thiruvenkatarajan et al. [15].

**Table 1 jcm-11-03860-t001:** Study characteristics.

Authors, Year	Country	Sources of Funding	Population	Study Size	Age (Years) Mean ± SD or Median (IQR)	Male Gender (%)	BMI (kg/m^2^) mean ± SD or Median (IQR)	Procedure	HFNC Device	Dosage of Sedatives (Initial Dose + Maintenance Dose)/Depth of Sedation	OSA or Snoring History (%)	ASA III or IV (%)	HFNC Setting		Control Setting		Outcome
													Flow (L/min)	FiO_2_ (%)	Device	Flow (L/min)	
						*Studies which included patients who were at low risk for hypoxemia*											
Lin et al., 2019	China	Academic and Industrial	Adult outpatient	1994	H: 48 ± 18.86 C: 47 ± 18.84	41.3	H: 22.84 ± 3.06 C: 22.96 ± 3.23	EGD	Optiflow	1–2 mg/kg propofol IVP + 0.5 mg/kg propofol IVP as needed/moderate to deep	25.4	0	60	100	NC	2	1, 3, 4
Teng et al., 2019	Taiwan	Academic	Age 20–80 years ASA I–II	152	H: 46.65 ± 15.37 C: 51.56 ± 12.52 ^d^	39.5	H: 22.51 ± 4.19 C: 23.44 ± 3.58 ^d^	EGD	Optiflow	0.05 mg/kg midazolam and 0.2 mcg/kg alfentanil IVP + propofol TCI at plasma target 1 ug/mL/deep	19.7	0	30	100	NC or	NC: 5	1, 3, 4
					51.07 ± 11.96 ^e^		22.90 ± 3.58 ^e^								MAB	MAB:5	
						*Studies which included patients who were at moderate to high risk for hypoxemia*											
Riccio et al., 2019	USA	Academic and Industrial	Age 18–80 years BMI > 40 kg/m^2^	59	H: 54 ± 8 C: 59 ± 7	13.6	H: 48 ± 7 C: 49 ± 10	Colonoscopy	Comfort Flo	30–100 mg propofol IVP + 120–150 mcg/kg/min propofol IV cont’ infusion /moderate to deep	16.9	88.1	Up to 60	36 to 40	NC	4	1, 3, 4
Mazzeffi et al., 2021	USA	Academic	Age > 17 years AND receiving advanced EGD	262	H: 62 ± 13 C: 62 ± 15	60.3	H: 28.3 ± 6.5 C: 28.2 ± 6.2	EGD + RFA or CA, ERCP, EUS, or others	Vapotherm Precision Flow	No prespecified protocol (drug of choice limited to propofol, fentanyl and midazolam)/GA	12.2	ND	20	ND	NC	6	1, 2
Thiruvenkatarajan et al., 2021	Australia	Academic and Industrial	Age > 18 years AND any of the risk factors of hypoxemia ^b^	131	H: 69.1 ± 17.7 C: 65.5 ± 18.9	42.0	H: 30.0 ± 7.1 C: 28.2 ± 7.1	ERCP ^a^	Optiflow	Propofol TCI at plasma target 1.5–2 mcg/mL + propofol TCI at plasma target 1–4 mcg/mL and 0.5–1 mcg/kg fentanyl IV as needed/ND	24.4	100	30 to 60	100	NC + MG	NC: 4 MG: 4	1, 2, 3
Nay et al., 2021	France	Academic and Industrial	Age > 18 years AND any of the risk factors of hypoxemia ^c^	379	H: 64 (54, 71) C: 64 (55, 71)	54.1	H: 27.0 (23.9, 30.5) C: 26.55 (24.1, 30.1)	EGD, Colonoscopy, or both	Optiflow	No prespecified protocol (drug of choice for initial and maintenance sedation limited to propofol IV. BZD or opioids IV as needed for agitation)/GA	7.9	27.7	70	50	NC or FM or NPC	Flow to reach FiO_2_ of 50%	1, 3, 4
Kim et al., 2021	Korea	No	Age > 20 years AND undergoing ERCP under prone positioning	72	H: 65.3 ± 13.4 C: 67.3 ± 14.4	65.3	H: 23.1 ± 4.1 C: 22.1 ± 3.5	ERCP	Optiflow	0.5 mg/kg propofol and 1 mcg/kg fentanyl IVP + 30 mcg/kg/min propofol IV cont’ infusion /moderate to deep	2.8	44.4	50	100	NC	5	1, 2, 3, 4
Lee et al., 2021	Korea	Academic	Age ≥ 65 years	187	H: 78 ± 7 C: 79 ± 7	54.5	H: 22.86 ± 5.62 C: 23.58 ± 3.82	ERCP	Optiflow	0.5–1 mg/kg propofol IVP + 10–20 mg propofol IVP as needed or 2–6 mg/kg/h propofol cont’ infusion /deep	ND	16.6	50	50	NC	5	1, 4

ASA, American Society of Anesthesiologists physical status classification system; BMI, body mass index; BZD, Benzodiazepines; C, conventional oxygen therapy; CA, cauterization; EGD, esophagogastroduodenoscopy; ERCP, endoscopic retrograde cholangiopancreatography; EUS, endoscopic ultrasound; FM, face mask; IQR, interquartile range; GA, general anesthesia; H or HFNC, high-flow nasal cannula; MAB, mandibular advancement device; MG, mouth guard; NC, nasal cannula; ND, not documented; NPC, nasopharyngeal catheter; OSA, obstructive sleep apnea; RFA, radiofrequency ablation; SD, standard deviation; and TCI, target controlled infusion. ^a^ Procedure duration anticipated to be more than 15 min. ^b^ Risk factors: ASA ≥ 3, BMI > 30, OSA, STOP-Bang ≥ 3. ^c^ Risk factors: Age > 60 years, ASA ≥ 2, history of heart of lung disease, BMI ≥ 30, OSA, STOP-Bang ≥ 3. ^d^ Standard bite block. ^e^ Mandibular advancement bite block. Outcome of interest: 1. Incidence of hypoxia 2. Incidence of hypercapnia 3. Need for minor airway interventions, such as chin lift, jaw thrust, or nasal airway insertion 4. Duration of procedure. Lin et al. [11], Teng et al. [12], Riccio et al. [13], Mazzeffi et al. [14], Thiruvenkatarajan et al. [15], Nay et al. [16], Kim et al. [17], Lee et al. [18].

**Table 2 jcm-11-03860-t002:** Certainty of evidence: HFNC compared to COT for patients receiving gastrointestinal endoscopies.

Certainty Assessment							Summary of Findings				
Participants (Studies) Follow-up	Risk of Bias	Inconsistency	Indirectness	Imprecision	Publication Bias	Overall Certainty of Evidence	Study Event Rates (%)		Relative Effect (95% CI)	Anticipated Absolute Effects	
							With COT	With HFNC		Risk with COT	Risk Difference with HFNC
Incidence of hypoxemia											
3236 (8 RCTs)	serious ^a^	serious ^b^	not serious	serious ^c^	publication bias strongly suspected ^d^	⨁◯◯◯ Very low	224/1645 (13.6%)	60/1591 (3.8%)	RR 0.51 (0.24 to 1.09)	136 per 1000	67 fewer per 1000 (from 103 fewer to 12 more)
Incidence of hypoxemia in patients at moderate to high risk for hypoxemia											
1090 (6 RCTs)	serious ^a^	serious ^b^	not serious	not serious	publication bias strongly suspected ^d^	⨁◯◯◯ Very low	125/543 (23.0%)	59/547 (10.8%)	RR 0.54 (0.31 to 0.96)	230 per 1000	106 fewer per 1000 (from 159 fewer to 9 fewer)
Incidence of severe hypoxemia											
2766 (4 RCTs)	serious ^a^	not serious	not serious	not serious	publication bias strongly suspected ^d^	⨁⨁◯◯ Low	34/1384 (2.5%)	11/1382 (0.8%)	RR 0.38 (0.20 to 0.74)	25 per 1000	15 fewer per 1000 (from 20 fewer to 6 fewer)
Incidence of hypercapnia											
393 (2 RCTs)	not serious	not serious	not serious	very serious ^e^	None	⨁⨁◯◯ Low	69/196 (35.2%)	86/197 (43.7%)	RR 1.24 (0.97 to 1.58)	352 per 1000	84 more per 1000 (from 11 fewer to 204 more)
Need for minor airway interventions											
2827 (6 RCTs)	serious ^a^	very serious ^f^	not serious	very serious ^g^	None	⨁◯◯◯ Very low	345/1463 (23.6%)	53/1364 (3.9%)	RR 0.31 (0.08 to 1.22)	236 per 1000	163 fewer per 1000 (from 217 fewer to 52 more)
Duration of procedure											
2667 (5 RCTs)	serious ^a^	not serious	not serious	serious ^h^	publication bias strongly suspected ^d^	⨁◯◯◯ Very low	1360	1307	-		MD 0.12 higher (0.04 lower to 0.28 higher)

CI: confidence interval; MD: mean difference; and RR: risk ratio. Explanations. ^a^. Enrollment of some concern and high overall risk-of-bias RCTs. ^b^. I-square > 50%. ^c^. Insufficient sample size in trial sequential analysis. ^d^. Major asymmetry in Doi plot. ^e^. Wide range of 95% confidence interval, cross non-significant line, and false negative with insufficient sample size in trial sequential analysis. ^f^. I-square > 90%. ^g^. Very wide range of 95% confidence interval. ^h^. Cross non-significant line, and false negative with insufficient sample size in trial sequential analysis.

## Data Availability

Not applicable.

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
