# Peer review of "Should We Use High-Flow Nasal Cannula in Patients Receiving Gastrointestinal Endoscopies? Critical Appraisals through Updated Meta-Analyses with Multiple Methodologies and Depiction of Certainty of Evidence"

_jcm, 2022, doi:10.3390/jcm11133860_

Round 1

Reviewer 1 Report

Dear authors

 A very rigorous methodology has been applied.

Perhaps, the discussion could be improved a little by adjusting it well to the results obtained.

The conclusions are very important

Author Response

Thank you for your comments.

We agree with your comment that the discussion could be improved and be better connected to the result section. We have rewritten the discussion section and explained each of the findings accordingly. We have also modified the conclusion section.

Reviewer 2 Report

This is a very interesting paper. Just two more informations might be added: The type of sedation (which remedy used) which might have an effect on the results; and HFNC should be explained in more detail.

Since I am not familiar with details of metaanalyses, it might be worthwhile that an expert looks to the paper. 

Author Response

Thank you for your comments.

Point 1: The type of sedation (which remedy used) which might have an effect on the results

Response 1: We have added the type of sedations (types, initial and maintenance doses, IV push/continuous infusion, and depth of sedation) in table 1 as well as the result section. Since all studies used propofol and only one study had a difference of the total dose of propofol between HFNC group and conventional oxygen therapy group, we did not perform further statistical analysis.

Point 2: HFNC should be explained in more detail

Response 2: We have also expanded the discussion of HFNC’s physiology benefits in details in the discussion section.

Reviewer 3 Report

This meta-analysis  deals with major problem frequently encountered during endoscopic procedures, namely severe hypoxemia. The methodology included an extensive number of studies and the statistical analysis is very elaborated. However some data regarding sedation protocols from the analysed studies may be useful and if it is possible some statistical analysis concerning sedation.

Author Response

Point 1: However some data regarding sedation protocols from the analysed studies may be useful and if it is possible some statistical analysis concerning sedation.

Reply 1: Thank you for your comments. We have added the type of sedations (types, initial and maintenance doses, IV push/continuous infusion, and depth of sedation) in table 1 as well as the result section. Since all studies used propofol and only one study had a difference of the total dose of propofol between HFNC group and conventional oxygen therapy group, we did not perform further statistical analysis.

Reviewer 4 Report

The authors have made a meta analysis of HFNO vs. conventional oxygen therpy during sedations for gastrointestinal endoscopies. The topic itself is important

Major comments: 

- HFNO (high flow nasal oxygenation) should have been included as a search term, as this describes the technology itself

- I can't find an analysis on high flow settings and the incidence of complications. It has been well described that higher flows can lead to some kind of Venturi phenomenon that entrains environmental air (and hence mainly nitrogen) into the lungs, making the inspired oxygen fraction much lower than what the settings of the HFNO device itself are. 

- There is a recent, very similar meta analysis published, please see https://onlinelibrary.wiley.com/journal/14431661. You may wish to consider the results from these colleagues and make your manuscript more distinct from theirs. 

Minor comments: 

-I would change the title to make it more "catchy". 

-Your statistical analysis is very sophisticated but hard to follow for the reader. 

- I suggest adding some sort of "take home message" for the clinician. 

- Also, the discussion should be somewhat more "vibrant". It it about advantages and disadvantages of HFNO. This sometimes gets lost.  

- Line 58: HFNO devices may be able to provide higher oxygen flows (at our department, we use devices that can provide up to 100 L/min). 

Overall, at the moment, the paper is too technical for a journal titled "Journal of Clinical Medicine". I suggest revising it in these points to make it more readable and have a clinical point for the reader, otherwise you technically excellent work may not be of great interest for the clinician. 

Round 2

Reviewer 4 Report

Thank you very much for the quick, extensive revision of the manuscript according to my suggestions, and also for adding HFNO. My comments have all been addressed in a satisfactory way. The changes have significantly improved the manuscript. Also, the manuscript benefits from the new references.

For the flow discussion, there is a technical simulation (https://pubmed.ncbi.nlm.nih.gov/34620089/) that found that higher flows may generate a significant Venturi effect. This cannot be measured in real patients, but I think it it worth discussing it.

Thank you for improving your manuscript.  

---- 

For your information only:

I agree that most stand-alone HFNO devices can deliver flows of 60-70L max. At our department, we use Hamilton C-6 ICU respirators in the physician-manned workplaces where we use HFNO for (deep) sedation, as they have several safety benefits over the standalone HFNO devices. They can provide flows of 100L/min at an FiO2 of 1.0 with the newest software. That is impressive, but of course it is still unknown if these excessive flows have additional benefits. 
